# The *Ormdl* genes regulate the sphingolipid synthesis pathway to ensure proper myelination and neurologic function in mice

Benjamin A Clarke[1†], Saurav Majumder[1†], Hongling Zhu[1†], Y Terry Lee[1], Mari Kono[1], Cuiling Li[1], Caroline Khanna[1], Hailey Blain[1], Ronit Schwartz[1], Vienna L Huso[1], Colleen Byrnes[1], Galina Tuymetova[1], Teresa M Dunn[2], Maria L Allende[1]*, Richard L Proia[1]*

[1]Genetics of Development and Disease Branch, National Institute of Diabetes and Digestive and Kidney Diseases, National Institutes of Health, Bethesda, United States; [2]Department of Biochemistry, Uniformed Services University of the Health Sciences, Bethesda, United States

*For correspondence:
MariaA@intra.niddk.nih.gov
(MLA);
richardp@intra.niddk.nih.gov (RLP)

†These authors contributed equally to this work

Competing interests: The authors declare that no competing interests exist.

**Abstract** Sphingolipids are membrane and bioactive lipids that are required for many aspects of normal mammalian development and physiology. However, the importance of the regulatory mechanisms that control sphingolipid levels in these processes is not well understood. The mammalian ORMDL proteins (ORMDL1, 2 and 3) mediate feedback inhibition of the de novo synthesis pathway of sphingolipids by inhibiting serine palmitoyl transferase in response to elevated ceramide levels. To understand the function of ORMDL proteins in vivo, we studied mouse knockouts (KOs) of the *Ormdl* genes. We found that *Ormdl1 and Ormdl3* function redundantly to suppress the levels of bioactive sphingolipid metabolites during myelination of the sciatic nerve. Without proper ORMDL-mediated regulation of sphingolipid synthesis, severe dysmyelination results. Our data indicate that the *Ormdls* function to restrain sphingolipid metabolism in order to limit levels of dangerous metabolic intermediates that can interfere with essential physiological processes such as myelination.

## Introduction

The sphingolipid metabolic pathway is a fundamental feature of all eukaryotic cells (*Merrill, 2011*). It is required to produce complex sphingolipids, such as sphingomyelin and the expansive glycosphingolipid family, that are plasma-membrane building blocks. It also generates bioactive metabolites (such as ceramide, sphingosine, and sphingosine-1-phosphate) that alter cell activities, including growth regulation and apoptosis, through interactions with receptors and enzymes. Some segments of the pathway have been specialized in a cell- and tissue-specific manner to supply essential sphingolipids that have unique properties needed for key physiological functions (e.g., the production and transport of ultra-long chain ceramides for creating the skin permeability barrier, or specific glycosphingolipids for forming the tightly packed myelin membrane that insulates neuronal axons) (*Dunn et al., 2019*).

The de novo sphingolipid biosynthetic pathway begins with the condensation of an amino acid (usually serine) and a fatty acyl-CoA (usually palmitoyl-CoA) by the serine palmitoyltransferase (SPT) enzyme complex (*Figure 1A*) (*Merrill, 2011*). The first product, 3-keto-dihydrosphingosine, is then reduced to form dihydrosphingosine, which is subsequently acylated with fatty acids of different chain lengths to produce dihydroceramide. Introduction of a double bond into the

**Figure 1.** *Ormdl3*, but not *Ormdl1* or *Ormdl2,* single knockout (KO) mice exhibit significantly increased levels of sphingolipids in the brain. (**A**) Schematic of the de novo sphingolipid biosynthetic pathway and its feedback inhibition by ORMDLs through the sensing of ceramide levels. SPT, serine palmitoyltransferase; 3KDHSph, 3-keto-dihydrosphingosine; DHSph, dihydrosphingosine; DHCer, dihydroceramide; Cer, ceramide; Sph, sphingosine; S1P, sphingosine-1-phosphate. (**B–D**) Generation of *Ormdl* KO mice. Panels show the intron-exon organizations of the *Ormdl* genes and the protein coding regions (white). (**B, C**) *Ormdl1* and *Ormdl2* KO mice were produced by CRISPR/Cas9-induced mutations, resulting in frameshifts and premature stop codons. The locations of sgRNA sequences (red), PAM sites (green), as well as the changes in DNA and protein are indicated. The base insertion in the CRISPR/Cas9 modified *Ormdl2* gene is underlined. (**D**) *Ormdl3* KO mice were generated by germline Cre-LoxP recombination to excise exons 2, 3, and part of exon 4, resulting in the deletion of the entire protein-coding sequence. (**E**) RT-qPCR of *Ormdl* WT RNA in brain of *Ormdl* KO mice relative to that in WT mice. The mice were 8 weeks old. Probes detect the WT *Ormdl* sequences. Data are expressed as means ± SD. Unpaired Student's *t* test; ***p<0.001. nd, not detectable. n = 4 for all genotypes. (**F**) Levels of dihydrosphingosine, total dihydroceramide, total ceramide, and sphingosine were determined by HPLC-tandem MS on lipid extracts of whole brains harvested from 8-week-old WT, *Ormdl1* KO, *Ormdl2* KO, *Ormdl3* KO, *Ormdl1/2* double KO, *Ormdl1/3* double KO, and *Ormdl2/3* double KO mice (*Figure 1—source data 1*). Data are expressed as means ± SD. One-way ANOVA with Bonferroni correction; *p<0.05, ***p<0.001. n = 8 for all genotypes. DKO, double knockout.

The online version of this article includes the following source data and figure supplement(s) for figure 1:

**Source data 1.** Levels of dihydrosphingosine, total dihydroceramide, total ceramide, and sphingosine from brains of WT, *Ormdl1* KO, *Ormdl2* KO, *Ormdl3* KO, *Ormdl1/2* double KO, *Ormdl1/3* double KO, and *Ormdl2/3* double KO mice.
**Figure supplement 1.** Generation of floxed *Ormdl3* mice.
**Figure supplement 2.** Levels of brain ceramide and dihydroceramide subspecies in *Ormdl* KO mice.
**Figure supplement 2—source data 1.** Levels of individual ceramide and dihydroceramide subspecies with different fatty-acid chain lengths from brains of WT, *Ormdl1* KO, *Ormdl2* KO, *Ormdl3* KO, *Ormdl1/2* double KO, *Ormdl1/3* double KO, and *Ormdl2/3* double KO mice.

sphingoid base generates ceramide, a central metabolic intermediate in the pathway. Ceramide, and its degradation product sphingosine, are bioactive and can cause cell death at elevated levels (*Hannun and Obeid, 2018*). Complex sphingolipids – sphingomyelin and glycosphingolipids – are generated by addition of hydrophilic head groups to the ceramide anchor.

Like sterols and glycerolipids, cellular sphingolipid levels are tightly regulated (*Breslow and Weissman, 2010*; *Brown et al., 2018*; *Harayama and Riezman, 2018*). The discovery that the *Ormdl* gene family mediates feedback inhibition of de novo sphingolipid synthesis has provided insight into a homeostatic mechanism that controls sphingolipid generation in mammals (*Breslow et al., 2010*; *Davis et al., 2019*). Three mammalian *Ormdl* genes exist (*Ormdl1*, *Ormdl2*, and *Ormdl3*), and they encode small transmembrane endoplasmic reticulum proteins with amino-acid identities of around 80% (*Hjelmqvist et al., 2002*). The ORMDL proteins, in complex with SPT, have the capacity to sense elevated ceramide levels and to inhibit SPT enzymatic activity, thereby blocking entry of de novo synthesized sphingolipid substrate into the sphingolipid biosynthetic pathway (*Davis et al., 2019*; *Han et al., 2019*; *Siow et al., 2015*) (*Figure 1A*).

The physiological contexts in which the ORMDLs are required are not well understood, but could inform us about when and why the de novo sphingolipid biosynthetic pathway requires negative regulatory control. Here, we investigated this issue by establishing knockout (KO) mice for each of the *Ormdl* genes. While the single KO mice were without overt phenotypes, we found that *Ormdl1/3* double KO mice exhibited a conspicuous phenotype, with elevated levels of sphingolipid metabolites, severe myelination defects and neurologic abnormalities. The results indicate that ORMDLs, functioning redundantly, are essential for maintaining control of the de novo sphingolipid biosynthetic pathway in a specific physiological context – myelination – where there is a high demand for sphingolipids.

## Results

### *Ormdl3*, but not *Ormdl1* or *Ormdl2*, single KO mice exhibit significantly increased levels of sphingolipids in brain

In order to identify physiologic functions that are associated with the *Ormdl* genes, we established whole-body *Ormdl1* KO, *Ormdl2* KO, and *Ormdl3* KO mice (*Figure 1B–D*, *Figure 1—figure supplement 1*). *Ormdl1* and *Ormdl2* KOs were created by CRISPR/Cas9 genome editing in embryos (*Wang et al., 2013*). In both cases, small deletions were introduced into exon 2, the first protein-coding exon of the two genes, to cause frameshifts in the coding sequences. These frameshifts produced premature terminations that eliminated more than 80% of the native protein-coding sequences (*Figure 1B and C*). For *Ormdl3*, gene targeting using homologous recombination at the *Ormdl3* locus in embryonic stem cells was applied. This introduced LoxP sequences flanking the region encompassed by *Ormdl3* exons 2 to 4, which contains the entire protein-coding region (*Figure 1—figure supplement 1*). Mice carrying the floxed *Ormdl3* allele were bred with mice expressing Cre recombinase under the control of the ubiquitous EIIA promoter element (*Lakso et al., 1996*) to generate mice with a germline deletion of the entire *Ormdl3* protein-coding region (*Ormdl3* KO) (*Figure 1D*). For each of the single-gene KOs, mice carrying homozygous mutant alleles were obtained. RT-qPCR mRNA expression assays using brain RNA with probes corresponding to the deleted coding sequences indicated a deficiency of the individual wild-type (WT) *Ormdl* mRNA expression in the corresponding KO mouse, consistent with the introduced genomic changes (*Figure 1E*).

We measured levels of dihydrosphingosine and dihydroceramide, two intermediates generated early in the sphingolipid biosynthetic pathway exclusively through de novo synthesis, as well as levels of ceramide and sphingosine, two bioactive metabolites generated through both de novo synthesis and recycling (*Merrill, 2011*) (*Figure 1A*), in the brains of the *Ormdl1* KO, *Ormdl2* KO, and *Ormdl3* KO mice (*Figure 1F*). The *Ormdl1* KO and *Ormdl2* KO mice had levels of these four types of sphingolipids that were generally similar to those measured in WT mice; however, the *Ormdl3* KO mice had significantly elevated brain levels of each of these sphingolipids compared with WT mice. This result suggests that amongst the *Ormdl*s, *Ormdl3* has the greatest influence on the levels of these sphingolipids, including those generated solely through de novo synthesis, consistent with ORMDL inhibitory action on SPT (*Figure 1A*).

## Elevated sphingolipids, neurologic phenotype and reduced viability when multiple *Ormdls* are deleted

*Ormdl1* KO, *Ormdl2* KO, and *Ormdl3* KO mice appeared overtly normal, with body weights not significantly different from those of WT mice at 8 weeks of age (*Figure 2A*). Intercross mating between respective single *Ormdl* KO mice produced litter sizes at weaning similar to those obtained through WT matings, suggesting normal fecundity and perinatal survival of *Ormdl1* KO, *Ormdl2* KO, and *Ormdl3* KO mice (*Figure 2B*).

To determine whether functional redundancy of the *Ormdls* may have masked the expression of a phenotype in the single *Ormdl* KO mice, we cross-bred mice to produce *Ormdl* double KO mice in each of the three possible combinations – *Ormdl1/2* double KO, *Ormdl1/3* double KO, and *Ormdl2/3* double KO mice – and then characterized those offspring.

As with the *Ormdl* single KO mice, dihydrosphingosine, dihydroceramide, ceramide and sphingosine levels were measured in the brains of the *Ormdl* double KO mice (*Figure 1F*). Total amounts of these four sphingolipid groups were not increased above WT levels in *Ormdl1/2* double KO brain (*Figure 1F*). However, levels of dihydrosphingosine, dihydroceramide, ceramide and sphingosine in *Ormdl1/3* double KO brain substantially exceeded those of the *Ormdl3* single KO mice (*Figure 1F*). *Ormdl2/3* double KO brain dihydrosphingosine and dihydroceramide levels were significantly elevated above WT levels, but not to the extent observed in *Ormdl1/3* double KO mouse brain (*Figure 1F*, *Figure 1—figure supplement 2*).

At 8 weeks of age, *Ormdl1/2* and *Ormdl2/3* double KO males and females had body weights similar to those of WT mice (*Figure 2A*). However, both male and female *Ormdl1/3* double KO mice weighed significantly less than WT mice. *Ormdl1/2* double KO and *Ormdl2/3* double KO mice were fertile and, when intercrossed, produced litter sizes similar to those of WT mating pairs (*Figure 2B*). Owing to their compromised physical and neurological status (below), breeding of *Ormdl1/3* double KO mice was not attempted.

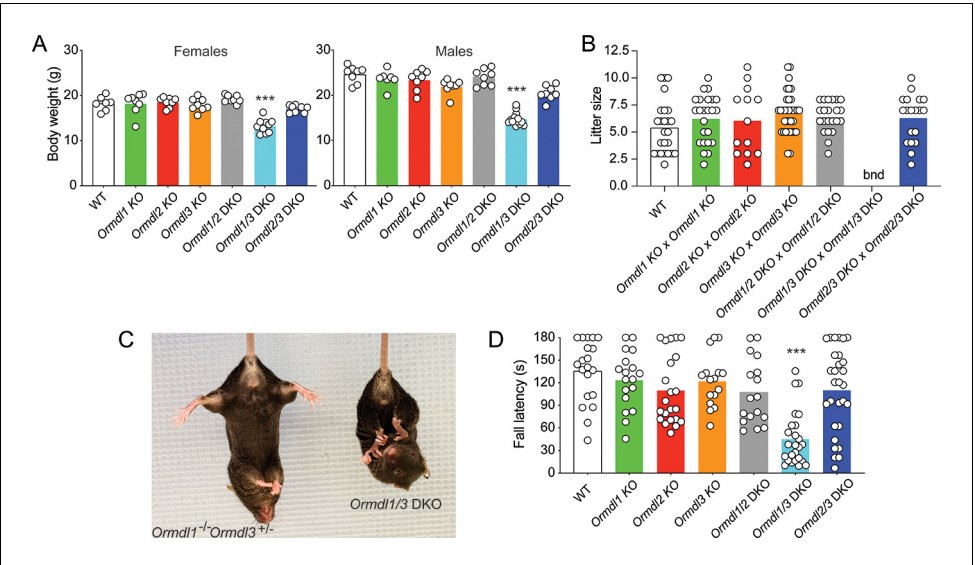

**Figure 2.** Elevated sphingolipids, neurologic phenotype and reduced viability when multiple *Ormdls* are deleted. (A) Body weight of 8-week-old male (n = 7–14) and female mice (n = 7–10) of the indicated genotypes. Each circle represents the weight of an individual mouse. Unpaired Student's *t* test; ***p<0.001 versus WT mice. (B) Litter size was determined at weaning for offspring of the indicated matings. Each circle represents the number of weanlings from an individual litter (n = 19–25). bnd, breeding not done. One-way ANOVA with Bonferroni correction. (C) Image of an 8-week-old *Ormdl1/3* double KO mouse (right) showing characteristic clasping of the hindlimbs upon tail suspension, a sign of neurodegeneration. An *Ormdl1⁻/⁻ Ormdl3⁺/⁻* littermate is shown (left). (D) Wire hang behavioral test. Eight-week-old mice were allowed to hang from a suspended wire using the forelimbs and the latency time to fall was recorded. The maximum hanging time was 180 s. Circles represent the mean of three determinations for each mouse (n = 16–29 mice for each genotype). One-way ANOVA with Bonferroni correction; ***p<0.001 versus WT mice. DKO, double knockout.

The *Ormdl1/3* double KO mice exhibited abnormal hindlimb clasping upon tail suspension, a sign of possible neurodegeneration (*Figure 2C*) (*Lalonde and Strazielle, 2011*). We used the forelimb wire hang behavioral test (*Aartsma-Rus and van Putten, 2014*), which measures strength and coordination, to assess the neuromuscular status of the single and double KO *Ormdl* mice at 8 weeks of age. The performance of *Ormdl1*, *Ormdl2*, and *Ormdl3* single KO mice and of *Ormdl1/2* and *Ormdl2/3* double KO mice was not significantly different from that observed for the WT mice (*Figure 2D*). However, the latency time to fall from the wire was significantly shorter for *Ormdl1/3* double KO mice than for WT mice, indicating that *Ormdl1/3* double KO mice had a behavioral deficit related to limb strength and coordination.

To determine the phenotypic consequences when more than one *Ormdl* gene was deleted, we crossed *Ormdl 1⁻/⁻ 2 ⁺/⁻ 3 ⁺/⁻* mice and determined the genotype of the pups produced at weaning (*Table 1*). The expected distribution of genotypes of offspring from these mating pairs, based on Mendelian considerations, was significantly different from the genotype distribution actually obtained from the 156 offspring (p<0.0001, Chi-square analysis). Notably, no triple *Ormdl1/2/3* KO mice were identified, although eight were predicted from the total number of offspring produced. Moreover, we also obtained much lower than predicted numbers of mice carrying only one WT *Ormdl* allele; that is, although 17 mice each were predicted for the *Ormdl 1⁻/⁻ 2⁻/⁻ 3⁺/⁻* and *Ormdl 1⁻/⁻ 2⁺/⁻ 3⁻/⁻* genotypes, only 5 and 0, respectively, were actually obtained. The results suggest that the absence of all six WT *Ormdl* alleles causes either embryonic or neonatal lethality prior to weaning. Supporting this conclusion is the observation that inheritance of only one WT *Ormdl2* or *Ormdl3* allele substantially reduced viability.

## Myelination is disrupted in *Ormdl1/3* double KO mice

Sphingolipids are major components of the lipid-rich myelin membrane that surrounds axons in the nervous system and are essential for proper myelin function (*Coetzee et al., 1996*). We hypothesized that the ORMDLs may be especially critical when sphingolipid synthesis demand is high, such as during the formation of myelin membranes after birth. If de novo sphingolipid synthesis is poorly regulated under these circumstances by the absence of ORMDLs, increased amounts of sphingolipids might be generated, thereby interfering with orderly myelination in *Ormdl1/3* double KO mice. To evaluate the impacts of the ORMDLs on myelination, we examined sciatic nerves at 6 weeks of age in WT, *Ormdl1* KO, *Ormdl3* KO, and *Ormdl1/3* double KO mice by transmission electron microscopy (EM) (*Figure 3A*). *Ormdl1/3* double KO mouse sciatic nerve exhibited a highly abnormal morphology when compared with sciatic nerve obtained from WT or single *Ormdl1* and *Ormdl3* KO mice. The nerves from the *Ormdl1/3* double KO mice had a significantly lower frequency of myelinated axons per field compared with WT sciatic nerves (*Figure 3B*). No significant difference in the numbers of myelinated axons was observed between sciatic nerves from WT and *Ormdl1* or *Ormd3* single KO mice. A significantly higher frequency of redundant myelin, generally appearing as myelin 'outfoldings' (*Golan et al., 2013*), was also observed in sciatic nerves from *Ormdl1/3* double KO mice compared with WT mice (*Figure 3C and D*).

**Table 1.** Analysis of offspring from *Ormdl1⁻/⁻ Ormdl2⁺/⁻ Ormdl3⁺/⁻* intercrosses.
Mouse genotypes were determined by PCR of tail-snip DNA of 156 pups at weaning from 38 litters derived from *Ormdl1⁻/⁻ Ormdl2⁺/⁻ Ormdl3⁺/⁻* intercrosses. The genotype distribution frequency of offspring, predicted by Mendelian considerations and actually obtained, is shown. The Chi-square test was used to determine whether the obtained distribution of genotypes was statistically different from the predicted Mendelian ratios; p<0.0001.

| | Ormdl1⁻/⁻ Ormdl2⁺/⁺ Ormdl3⁺/⁻ | Ormdl1⁻/⁻ Ormdl2⁺/⁺ Ormdl3⁺/⁺ | Ormdl1⁻/⁻ Ormdl2⁺/⁺ Ormdl3⁻/⁻ | Ormdl1⁻/⁻ Ormdl2⁺/⁻ Ormdl3⁺/⁻ | Ormdl1⁻/⁻ Ormdl2⁺/⁻ Ormdl3⁺/⁺ | Ormdl1⁻/⁻ Ormdl2⁺/⁻ Ormdl3⁻/⁻ | Ormdl1⁻/⁻ Ormdl2⁻/⁻ Ormdl3⁺/⁺ | Ormdl1⁻/⁻ Ormdl2⁻/⁻ Ormdl3⁺/⁻ | Ormdl1⁻/⁻ Ormdl2⁻/⁻ Ormdl3⁻/⁻ |
|---|---|---|---|---|---|---|---|---|---|
| Observed # | 16 | 20 | 9 | 33 | 62 | 0 | 11 | 5 | 0 |
| Observed % | 10.26 | 12.82 | 5.77 | 21.15 | 39.74 | 0 | 7.05 | 3.21 | 0 |
| Predicted # | 9.75 | 19.5 | 9.75 | 19.5 | 39 | 19.5 | 9.75 | 19.5 | 9.75 |
| Predicted % | 6.25 | 12.5 | 6.25 | 12.5 | 25 | 12.5 | 6.25 | 12.5 | 6.25 |

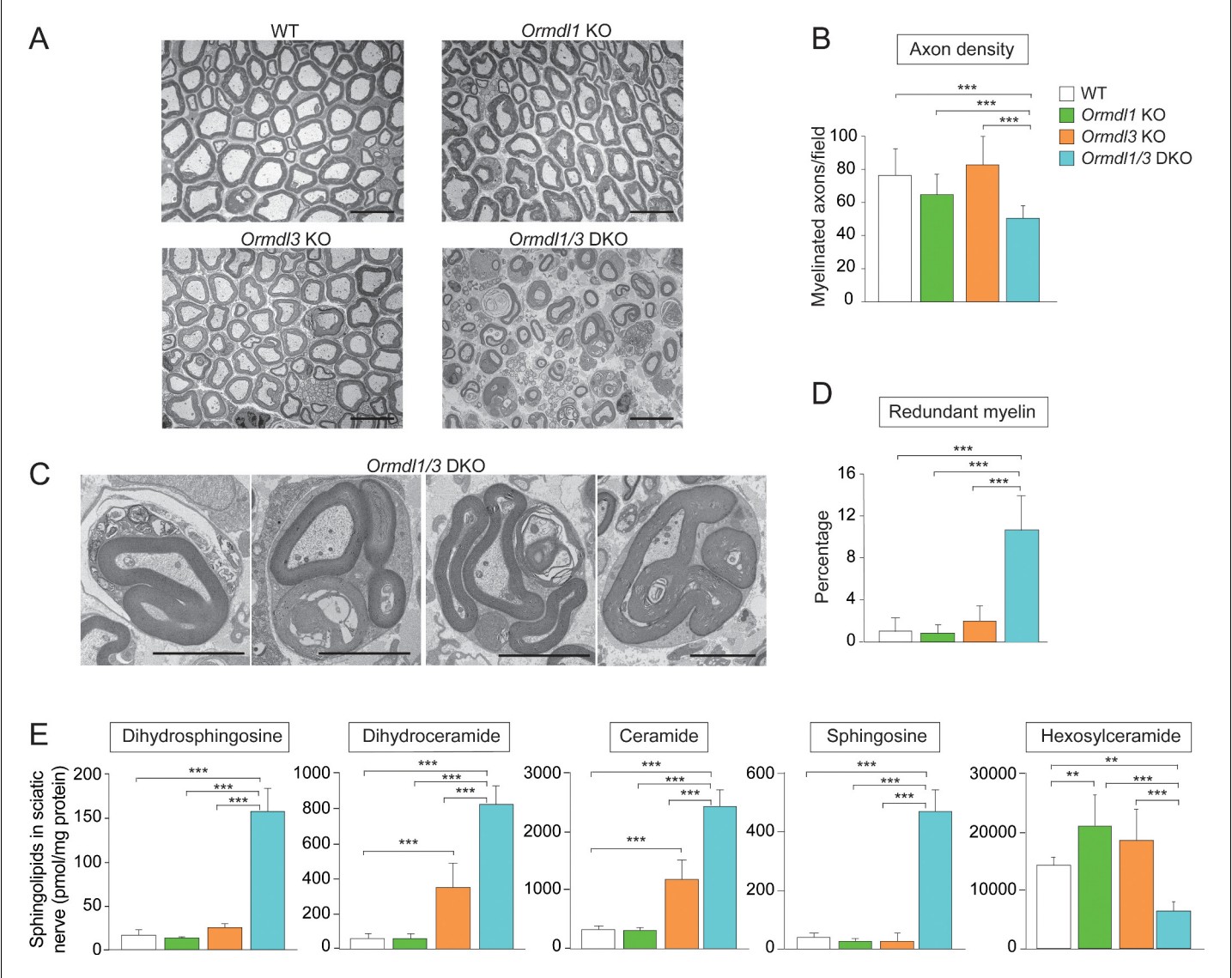

**Figure 3.** Myelination is disrupted in *Ormdl1/3* double KO mice. (**A**) Representative transmission EM images of sciatic nerve of 6-week-old WT, *Ormdl1* KO, *Ormdl3* KO, and *Ormdl1/3* double KO mice. Scale bars, 10 µm. (**B**) Axon density in sciatic nerve of 6-week-old WT, *Ormdl1* KO, *Ormdl3* KO, and *Ormdl1/3* double KO mice was determined by quantifying the number of myelinated axons in 7–12 EM fields per genotype. Data are expressed as means ± SD. One-way ANOVA with Bonferroni correction; ***p<0.001. n = 3 for WT, n = 2 for *Ormdl1* KO, n = 2 for *Ormdl3* KO, n = 3 for *Ormdl1/3* double KO mice. (**C**) Example images of redundant myelin figures in sciatic nerve axons of 6-week-old *Ormdl1/3* double KO mice. Scale bars, 5 µm. (**D**) Percentage of myelinated axons in sciatic nerve of 6-week-old WT, *Ormdl1* KO, *Ormdl3* KO, and *Ormdl1/3* double KO mice showing redundant myelination was quantified in 7–12 EM fields per genotype. Data are expressed as means ± SD. One-way ANOVA with Bonferroni correction; ***p<0.001. n = 3 for WT, n = 2 for *Ormdl1* KO, n = 2 for *Ormdl3* KO, n = 3 for *Ormdl1/3* double KO mice. (**E**) Levels of dihydrosphingosine, total dihydroceramide, total ceramide, sphingosine and hexosylceramide were determined by HPLC-tandem MS on lipid extracts of sciatic nerve from 8-week-old WT, *Ormdl1* KO, *Ormdl3* KO, and *Ormdl1/3* double KO mice (*Figure 3—source data 1*). Data are expressed as means ± SD. One-way ANOVA with Bonferroni correction; ***p<0.001. n = 8 for all genotypes. DKO, double knockout.

The online version of this article includes the following source data and figure supplement(s) for figure 3:

**Source data 1.** Levels of dihydrosphingosine, total dihydroceramide, total ceramide, sphingosine and total hexosylceramide from sciatic nerves of WT, *Ormdl1* KO, *Ormdl3* KO, and *Ormdl1/3* double KO mice.

**Figure supplement 1.** Levels of sciatic nerve ceramide, dihydroceramide and hexosylceramide subspecies in *Ormdl1* KO, *Ormdl3* KO, and *Ormdl1/3* double KO mice.

**Figure supplement 1—source data 1.** Levels of individual ceramide, dihydroceramide and hexosylceramide subspecies with different fatty-acid chain lengths from sciatic nerves of WT, *Ormdl1* KO, *Ormdl3* KO, and *Ormdl1/3* double KO mice.

We measured levels of sphingolipid metabolites, dihydrosphingosine, dihydroceramide, ceramide, and sphingosine in sciatic nerves from these same genotypes. Sciatic nerve of *Ormdl1* KO mice had levels of these sphingolipids that were not significantly different from those from WT mice, whereas sciatic nerve of *Ormdl3* KO mice had significantly elevated levels of both dihydroceramide and ceramide compared with nerve from WT mice. However, the *Ormdl1/3* double KO sciatic nerve had significantly elevated levels of all four of these sphingolipid species compared with levels observed in the *Ormdl3* KO mice (*Figure 3E*).

We also measured sciatic nerve levels of hexosylceramide, comprising glucosyl and galactosylceramide, the major glycosphingolipids in myelin (*Wattenberg, 2019*). Hexosylceramide levels in the *Ormdl1* and *Ormdl3* single KO sciatic nerve were significantly increased compared to those in WT sciatic nerve (*Figure 3E*, *Figure 3—figure supplement 1C*). However, sciatic nerve from *Ormdl1/3* double KO mice had decreased hexosylceramide levels, even though they had an 8-fold increase in the level of ceramide, the direct precursor for hexosylceramide synthesis. This result suggests that the large increase in precursor sphingolipid substrate ceramide may have saturated the terminal glycosylation step for hexosylceramide production.

## Increased de novo sphingolipid biosynthesis in myelin-producing cells mimicked the phenotype of *Ormdl1/3* KO mice

Our results are consistent with the conclusion that elevated sphingolipid synthesis resulting from *Ormdl1/3* deficiency causes dysmyelination and motor-function abnormalities in mice. Although reductions in the synthesis of specific sphingolipids have been associated with defects in myelination (*Coetzee et al., 1996*), it has not been shown previously that increased de novo sphingolipid synthesis can cause myelination defects. To address this issue, we constructed a mouse model that allows for conditional expression of SPT activity (*Figure 4—figure supplement 1A*). The transgenic mouse carries a single-chain version of the SPT enzyme with its three core subunits (SPTLC2, SPTSSA and SPTLC1) genetically fused (fusion[f]SPT) and under the control of a promoter that is activated by Cre recombinase expression (Stop-fSPT) (*Alexaki et al., 2014*; *Gable et al., 2010*). The Stop-fSPT cassette was introduced into the *Rosa26* locus using the phiC31 integrase system (*Tasic et al., 2011*).

In order to validate the model, we first generated mice carrying Stop-fSPT along with the *Mx1-Cre* gene, which can be activated by polyinosinic-polycytidylic acid (pIpC) administration to induce Cre expression in the liver (*Kühn et al., 1995*). After administration of pIpC, the fSPT polypeptide was detected by Western blotting, along with significantly increased SPT enzymatic activity, in Stop-fSPT/*Mx1-Cre* mouse liver compared with the liver of mouse controls carrying only Stop-fSPT (*Figure 4—figure supplement 1B and C*). Ceramide levels were significantly increased in the liver of mice expressing fSPT compared with that of control mice (*Figure 4—figure supplement 1D*).

Next, we specifically expressed the fSPT gene in cells that produce myelin in order to determine whether elevated sphingolipid synthesis causes dysmyelination, as was observed in the *Ormdl1/3* double KO mice. We generated mice carrying Stop-fSPT and the *Cre* gene under the control of the tamoxifen-inducible *Plp1* promoter (*Doerflinger et al., 2003*), which would allow for conditional fSPT expression in Schwann cells and oligodendrocytes upon *Cre* activation. *Cre* expression was induced at 4 weeks of age by use of a tamoxifen diet. After 2 weeks, the Stop-fSPT/*iPlp1-Cre* mice that had been fed the tamoxifen diet exhibited severe hindlimb paralysis (*Figure 4A*). Sciatic nerve expression of the fSPT protein was confirmed by Western blotting (*Figure 4B*). Transmission EM analysis of sciatic nerves from fSPT-overexpressing mice revealed reduced axon density (*Figure 4C and D*) and a significantly increased frequency of redundant myelin structures (*Figure 4E and F*), similar to those found in *Ormdl1/3* double KO mice (*Figure 3C and D*). Levels of dihydrosphingosine, total dihydroceramide, total ceramide, and sphingosine were significantly elevated in the sciatic nerves of the fSPT-expressing mice compared with those of control mice (*Figure 4G*, *Figure 4—figure supplement 2A and B*). However, as in the *Ormdl1/3* double KO mice, hexosylceramide levels were decreased in sciatic nerves of fSPT-expressing mice even though these mice had highly elevated ceramide levels (*Figure 4G – Figure 4—figure supplement 2C*).

## Discussion

The ORMDLs have been identified as regulators that mediate feedback inhibition of de novo sphingolipid synthesis to control sphingolipid levels (*Breslow et al., 2010*; *Davis et al., 2019*; *Hagen-*

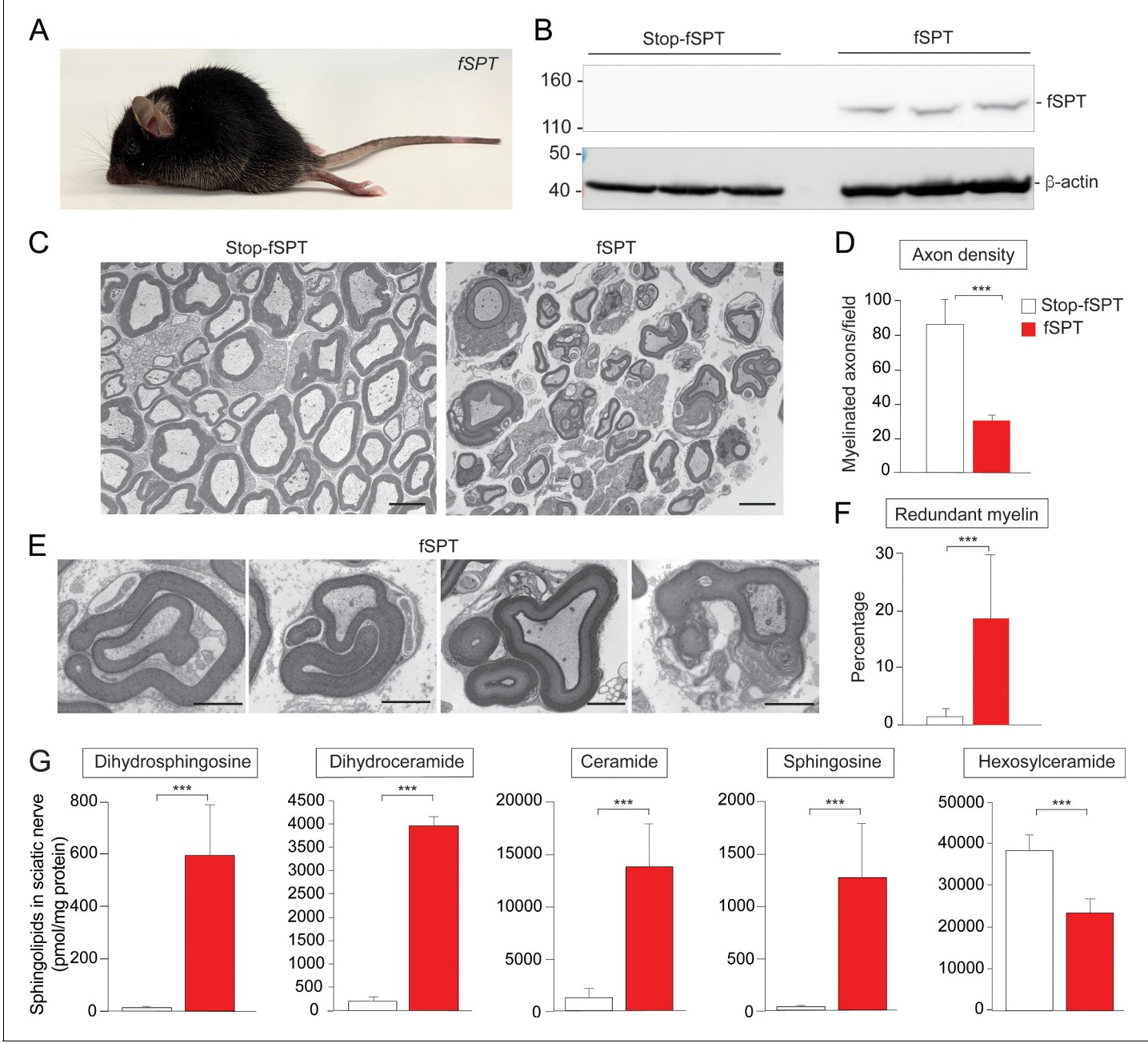

**Figure 4.** Increased de novo sphingolipid biosynthesis in myelin-producing cells mimicked the phenotype of *Ormdl1/3* KO mice. (**A**) Six-week-old fSPT/*iPlp1-Cre* (fSPT) mouse after treatment with tamoxifen for 2 weeks. (**B**) Western blot of fSPT expression in sciatic nerve of 6-week-old mice carrying Stop-fSPT without and with iPlp1-*Cre* (fSPT) after treatment with tamoxifen for 2 weeks. Bottom panel represents the same blot reprobed with an antibody against β-actin as a loading control. n = 3 for both genotypes. (**C**) Representative transmission EM images of sciatic nerve of 6-week-old Stop-fSPT and Stop-fSPT/*iPlp1-Cre* (fSPT)mice after treatment with tamoxifen for 2 weeks. Scale bars, 5 μm. (**D**) Axon density in sciatic nerve of 6-week-old Stop-fSPT and fSPT/*iPlp1-Cre* (fSPT) mice after treatment with tamoxifen for 2 weeks was determined by quantifying the number of myelinated axons in five and six EM fields for Stop-fSPT and fSPT/*iPlp1-Cre* mice (fSPT), respectively. Data are expressed as means ± SD. One-way ANOVA with Bonferroni correction; ***p<0.001. n = 2 for both genotypes. (**E**) Example images of redundant myelin figures in sciatic nerve of 6-week-old fSPT/*iPlp1-Cre* (fSPT) mice after treatment with tamoxifen for 2 weeks. Scale bars, 2 μm. (**F**) Percentage of myelinated axons showing redundant myelination in sciatic nerve of 6-week-old Stop-fSPT and fSPT/*iPlp1-Cre* (fSPT) mice after treatment with tamoxifen for 2 weeks, quantified in five and six EM fields for Stop-fSPT and fSPT mice, respectively. Data are expressed as means ± SD. One-way ANOVA with Bonferroni correction; ***p<0.001. n = 2 for both genotypes. (**G**) Levels of dihydrosphingosine, total dihydroceramide, total ceramide, sphingosine and hexosylceramide were determined by HPLC-tandem MS on lipid extracts of sciatic nerve from 6-week-old Stop-fSPT and fSPT/*iPlp1-Cre* (fSPT) mice after treatment with tamoxifen for 2 weeks (***Figure 4—source data***

*Figure 4 continued on next page*

*Figure 4 continued*

*1*). Data are expressed as means ± SD. One-way ANOVA with Bonferroni correction; ***p<0.001. n = 5 for Stop-fSPT, n = 7 for fSPT/*iPlp1-Cre* (fSPT) mice after treatment with tamoxifen for 2 weeks.

The online version of this article includes the following source data and figure supplement(s) for figure 4:

**Source data 1.** Levels of dihydrosphingosine, total dihydroceramide, total ceramide, sphingosine and hexosylceramide from sciatic nerves of mice carrying Stop-fSPT, without or with *iPlp1-Cre* (fSPT), after treatment with tamoxifen.

**Figure supplement 1.** Generation and characterization of fSPT conditional mutant mice.

**Figure supplement 1—source data 1.** Levels of individual ceramide subspecies with different fatty-acid chain lengths, C16-dihydroceramide, dihydrosphingosine, total ceramide, and sphingosine from liver of mice carrying Stop-fSPT, without and with *Mx1-Cre* (fSPT), after treatment with pIpC.

**Figure supplement 2.** Levels of sciatic nerve ceramide and dihydroceramide subspecies in mice overexpressing fSPT.

**Figure supplement 2—source data 1.** Levels of individual subspecies of ceramide, dihydroceramide and hexosylceramide with different fatty-acid chain lengths from sciatic nerves of mice carrying Stop-fSPT, without or with *iPlp1-Cre* (fSPT), after treatment with tamoxifen.

*Euteneuer et al., 2012*; *Zhakupova et al., 2016*). They are believed to act by sensing elevations in ceramide levels and by inhibiting SPT through direct protein–protein interactions within a multi-sub-unit enzyme complex (*Davis et al., 2019*), although elevated free and phosphorylated sphingoid bases have also been suggested to inhibit SPT via ORMDLs (*Hagen-Euteneuer et al., 2012*). The results presented here establish ORMDLs as functionally important modulators of in vivo sphingolipid levels in the nervous system. By evaluating mice in which either one or combinations of two of the three *Ormdl* genes were deleted, we found that mice lacking both *Ormdl1* and *Ormdl3* exhibit abnormal myelination in the sciatic nerve, as well as highly elevated sphingolipid levels in the nervous system. When de novo synthesis of sphingolipids was directly increased in myelin-producing cells by overexpression of fSPT, phenotypic manifestations similar to those seen in the *Ormdl1/3* double KO mice occurred. These findings indicate that ORMDLs suppress sphingolipid levels that, when elevated, cause dysmyelination.

After birth, during the acute phase of myelination, the Schwann cell membrane surface area expands several thousand times (*Webster, 1971*). The myelin membrane is approximately 70–80% lipid by weight, with much of the non-sterol lipid made up of sphingolipids (*Wattenberg, 2019*). Thus, the formation of the myelin membrane requires an extremely high influx of new substrate through the sphingolipid biosynthetic pathway to produce the mature sphingolipids, largely galacto-sylceramide and sulfated galactosylceramide, that are needed for proper myelin formation (*Coetzee et al., 1996*). Under conditions of very high sphingolipid synthesis, a lack of feedback inhibition could lead to high levels of ceramide if ceramide production exceeded the capacity of the downstream pathway to convert it into mature sphingolipids. Indeed, we found that the *Ormdl1/3* double KO mouse sciatic nerve contained high levels of ceramide together with lower levels of the mature sphingolipid hexosylceramide. The lower density of myelinated axons in the sciatic nerve of *Ormdl1/3* double KO mice may be a consequence of cell death due to this build-up of toxic sphingolipid precursor metabolites (*Hannun and Obeid, 2018*; *Riley and Merrill, 2019*). Our findings are consistent with the idea that a key physiological function of the ORMDLs is to keep de novo sphingolipid synthesis in check so that ceramide production does not exceed the metabolic capacity of the cell to convert ceramide into mature, non-toxic sphingolipids that are destined for physiological processes (*Davis et al., 2019*).

In this study, *Ormdl1/3* double KO mice exhibited a striking increase in the frequency of axons featuring redundant myelination. Redundant myelination, often observed as loops of myelin growing into the Schwann cell cytoplasm or surrounding the axon outside the normal myelin sheath, is believed to occur because of excessive myelin membrane growth or improper lipid composition (*Golan et al., 2013*). Unbalanced sphingolipid synthesis, caused either by deletion of the *Ormdl*s in the *Ormdl1/3* double KO mice or by the overexpression of fSPT in myelin-producing cells, may have promoted the formation of myelin with abnormal lipid composition through the excessive production of sphingolipid precursors relative to mature sphingolipids. Incorrect sphingolipid composition, caused by a deficiency of galactosyceramide, has been shown to produce redundant myelin profiles (*Dupree et al., 1998*). Charcot-Marie-Tooth disease, an inherited demyelinating neuropathy, often features redundant myelination as its prominent histologic feature (*Cotter et al., 2010*; *Goebbels et al., 2012*; *Golan et al., 2013*). Interestingly, a distinct Charcot-Marie-Tooth phenotype

in humans is caused by deficiency in the sphingolipid degradation enzyme, S1P lyase, a condition that can elevate ceramide levels (*Atkinson et al., 2017*; *Prasad et al., 2017*).

When we attempted to generate *Ormdl* triple KO mice by crossing *Ormdl* $1^{-/-}$ $2^{+/-}$ $3^{+/-}$ mice, no viable *Ormdl1/2/3* triple KO mice were identified at weaning. Further, the frequency of weaned pups with only one functional *Ormdl* allele produced from these crossings was much lower than that predicted from Mendelian considerations. These data suggest that the ORMDLs may also be needed for embryonic or neonatal viability. During embryonic growth, high levels of de novo sphingolipid synthesis would be required to support the extensive cellular proliferation that occurs (*Hojjati et al., 2005*). Immediately after birth, de novo sphingolipid synthesis is also elevated in epidermal cells to supply specialized sphingolipids in support of skin barrier formation (*Holleran et al., 2006*). The lack of negative feedback control on sphingolipid biosynthesis caused by the absence of ORMDLs in the *Ormdl* triple KO mice may have disrupted these or other critical developmental processes in which sphingolipids play an essential role.

Our results indicate that the individual ORMDLs can partially compensate for each other in both the control of sphingolipid levels and to prevent expression of severe phenotypes. Redundancy of ORMDL function may not be surprising given the extremely high sequence identity between the three ORMDL isoforms. The reason for redundancy of ORMDLs may reflect an organismal advantage in having robust control over a pathway that generates toxic metabolites but is also required under certain developmental scenarios to produce extremely high levels of specialized sphingolipid end-products. Under circumstances of high metabolic flux, small perturbations in downstream enzyme activities or transport mechanisms could result in catastrophic consequences if ceramide levels were allowed to increase. The existence of multiple isoforms may also reflect independent functions for the ORMDLs, such as differential responses to individual ceramide species according to their chain length, degree of saturation, or some other aspect of structure. Interestingly, in *Ormdl1/3* double KO brain, dihydroceramides with very long chain fatty acids (C22-26) did not increase, whereas in Ormdl1/3 double KO sciatic nerve, dihydroceramides with very long chain fatty acids were highly elevated. This result suggests that differences exist between these tissues, either in the flux of sphingolipid substrate through the pathway or in regulation by ORMDLs.

Our results emphasize the critical role that sphingolipid synthesis plays in the nervous system, as well as the balancing act that occurs within the sphingolipid biosynthesis pathway between the production of essential sphingolipids and the control of toxic metabolic intermediates. It is well known that genetic blocks in the sphingolipid metabolic pathway, especially in lysosomal degradation, cause severe neurodegeneration (*Dunn et al., 2019*). We have now shown that tight regulatory control of sphingolipid synthesis via the ORMDLs is also necessary to prevent nervous system damage during myelination, which relies on a large pool of mature sphingolipids.

# Materials and methods

**Key resources table**

| Reagent type (species) or resource | Designation | Source or reference | Identifiers | Additional information |
|---|---|---|---|---|
| Genetic reagent (*M. musculus*) | *Ormdl1* KO mice | This paper | RRID: IMSR_JAX:034646 | See 'Materials and methods' |
| Genetic reagent (*M. musculus*) | *Ormdl2* KO mice | This paper | RRID: IMSR_JAX:034645 | See 'Materials and methods' |
| Genetic reagent (*M. musculus*) | *Ormdl3* Floxed mice | This paper | RRID: IMSR_JAX:034644 | See 'Materials and methods' |
| Genetic reagent (*M. musculus*) | *EIIA-Cre* mice | Jackson Laboratory | RRID: IMSR_JAX:003724 | Mice expressing Cre transgene under the control *EIIa* promoter |
| Genetic reagent (*M. musculus*) | STOP-fSPT transgenic mice | This paper | | See 'Materials and methods' |

*Continued on next page*

Continued

| Reagent type (species) or resource | Designation | Source or reference | Identifiers | Additional information |
|---|---|---|---|---|
| Genetic reagent (*M. musculus*) | *PLP/creER^T* mice | Jackson Laboratory | RRID: IMSR_JAX:005975 | Mice expressing tamoxifen inducible-mouse *Plp1* promoter |
| Genetic reagent (*M. musculus*) | *Mx-Cre* mice | Jackson Laboratory | RRID: IMSR_JAX:003556 | Mice expressing Cre transgene under the control of the mouse *Mx1* promoter |
| Antibody | Monoclonal anti-human SPTLC1 | Santa Cruz Bio-technology | RRID:AB_10917035; clone H-1, cat. # sc-374143 | For western blot (1:1000 dilution) |
| Antibody | Anti-mouse IgG-HRP conjugated | Millipore | RRID:AB_9045; Cat. # AP-124P | For western blot (1:1000 dilution) |
| Antibody | Anti-mouse β-actin -HRP conjugated | Abcam | RRID:AB_8674; AC-15, cat # ab49900 | For western blot (1:100,000 dilution) |
| Sequence-based reagent | Ormdl1For | | PCR primer | Genotyping *Ormdl1* KO mice: 5'-TGTAATGAACAGCCGTGGTAT |
| Sequence-based reagent | Ormdl1Rev | | PCR primer | Genotyping *Ormdl1* KO mice: 5'-GCAGAAGGGGATGCTGAGTAATA |
| Sequence-based reagent | Ormdl2For | | PCR primer | Genotyping *Ormdl2* KO mice: 5'-CACATGCAGCAGTCCTACCA |
| Sequence-based reagent | Ormdl2Rev | | PCR primer | Genotyping *Ormdl2* KO mice: 5'-GTTGGACTCCTGCCTGATCC |
| Sequence-based reagent | NDEL1 | | PCR primer | Genotyping *Ormdl3* KO mice: 5'-GCAGGAGGAAGAGGCCCTCAG |
| Sequence-based reagent | NDEL2 | | PCR primer | Genotyping *Ormdl3* KO mice: 5'-CTCTTGACTGCCGCTCTGCAAAAGAG |
| Sequence-based reagent | SIAE4 | | PCR primer | Genotyping *Ormdl3* KO mice: 5'-CACGGCGCAGGGTTCTAATACATAC |
| Sequence-based reagent | ForSPT | | PCR primer | Genotyping fSPT mice: 5'-CATCGAGCTGAAGGGCATCG |
| Sequence-based reagent | RevSPT | | PCR primer | Genotyping fSPT mice: 5'-GTTATCTAGAATTATAGACGCGCTAG |
| Sequence-based reagent | ForR26 | | PCR primer | Genotyping fSP*T* mice: 5'-AGTTCTCTGCTGCCTCCTGGCTTCT |
| Sequence-based reagent | CreFor | | PCR primer | Genotyping fSPT mice: 5'-GCCTGCATTACCGGTCGATGC |
| Sequence-based reagent | CreRev | | PCR primer | Genotyping: 5'-CAGGGTGTTATAAGCAATCCC |
| Sequence-based reagent | TaqmanFWD | | *Ormdl1* taqman assay primer | *Ormdl1* taqman assay primer: 5'-ATGAATGTTGGAGTTGCCCAC |
| Sequence-based reagent | TaqmanREV | | *Ormdl1* taqman assay primer | *Ormdl1* taqman assay primer: 5'-GAACACTGCAGAAGGGGATG |

*Continued on next page*

*Continued*

| Reagent type (species) or resource | Designation | Source or reference | Identifiers | Additional information |
|---|---|---|---|---|
| Sequence-based reagent | *Ormdl1* probe | Applied Biosystems | Custom | *Ormdl1* taqman assay probe: 5'-TGTAGATGACCCAAAATGGT |
| Sequence-based reagent | *Ormdl2* assay-on-demand | Applied Biosystems | Mm00452481_g1 | |
| Sequence-based reagent | *Ormdl3* assay-on-demand | Applied Biosystems | Mm00787910_sH | |

## Mouse generation and genotyping

To obtain *Ormdl1* and *Ormdl2* KO alleles (*Figure 1B and C*), sgRNAs (Horizon, Cambridge, UK) corresponding to sequences in exon 2 downstream of the start codon of each gene, and the Cas9 protein (Horizon) were microinjected into C57BL/6 (Taconic Biosciences, Rensselaer, NY) embryos. The injected embryos were implanted into pseudo-pregnant surrogates (*Wang et al., 2013*). The offspring were screened by sequencing to identify mice carrying insertions or deletions in either the *Ormdl1* or the *Ormdl2* gene. For *Ormdl1*, the following primers were used to amplify the region edited by the corresponding sgRNA/Cas9 complex: 5'-GGCTAGAAAAACAAGCTTTGGAA-3'; 5'-TGGTGTCTCACTGCTTCCTTC-3'. The *Ormdl1* PCR fragment was sequenced using 5'-CAAGCTTTGGAAAAAGAAGCCA-3' as a primer. For *Ormdl2*, the following primers were used to amplify the region edited by the sgRNA/Cas9: 5'-GAACAGCTGGTGACTTTGTTTTGT-3'; 5'-GAGTAAAAACCCCACTTGTGTGAG-3'. The *Ormdl2* PCR fragment was sequenced using 5'-TGTTTTGTTTTGGTTGGGATAG-3' as a primer. Founder mice with the deletions shown in *Figure 1B and C* were backcrossed to C57BL/6 mice to derive mouse colonies carrying the *Ormdl1* and *Ormdl2* KO alleles.

To modify the *Ormdl3* locus, a targeting vector was prepared using a 10.2-kb genomic region containing exons 2 to 4 (*Figure 1—figure supplement 1*). The linearized targeting vector was transfected into BA1 (C57BL/6 × 129/SvEv) embryonic stem cells (Ingenious Targeting Laboratory, Ronkonkoma, NY). Correctly targeted embryonic stem cells were identified by PCR and Southern blotting. The neomycin cassette was removed by crossing F₁ mice to mice expressing FRT recombinase (Stock No: 009086, Jackson Laboratory, Bar Harbor, ME). *Ormdl3* KO mice were obtained by subsequent crossing with C57BL/6 mice expressing Cre recombinase under the EIIA promoter (Stock No: 003724, Jackson Laboratory) (*Lakso et al., 1996*) to excise the DNA between the LoxP sites, causing the deletion of the entire amino-acid-coding sequence for *Ormdl3* (*Figure 1D*). Mice were then backcrossed seven times to C57BL/6 mice to derive the *Ormdl3* mouse colonies.

To generate mice that enabled the conditional expression of fusion SPT (fSPT), the three individual SPT subunits (in the following order: SPTLC2, SPTSSA, and SPTLC1) were cloned as a single-chain coding unit (*Alexaki et al., 2014*; *Gable et al., 2010*) linked to EGFP by a T2A peptide sequence (*Figure 4—figure supplement 1A*). A stop cassette flanked by LoxP sites was inserted between the CAG promoter and the fSPT coding unit (*Niwa et al., 1991*). The expression cassette was inserted into the *Rosa26* locus using the TARGATT methodology (*Tasic et al., 2011*). fSPT transgenic mice were crossed with mice expressing Cre recombinase under the control of a tamoxifen-inducible-mouse *Plp1* promoter (Stock No: 005975, Jackson Laboratory) (*Doerflinger et al., 2003*) or a pIpC-inducible *Mx1* promoter (Stock No: 003556, Jackson Laboratory) (*Kühn et al., 1995*). fSPT/*iPlp1-Cre* mice were fed a tamoxifen diet (500 mg/kg, Envigo, Somerset, NJ) starting at 4 weeks of age to induce Cre-mediated excision of the stop cassette. fSPT/*Mx1-Cre* mice were injected intraperitonally 3 times every other day starting at 8 weeks of age with 150 µL of 2.5 mg/mL pIpC (Sigma, St. Louis, MO) in PBS to induce Cre-mediated removal of the stop cassette. fSPT/*Mx1-Cre* mice were analyzed 4 weeks after starting pIpC treatment. fSPT/*iPlp1-Cre* mice were analyzed for 2 weeks after treatment with tamoxifen diet.

Mice were genotyped by PCR of tail-snip DNA using the following primers and reactions:

*Ormdl1* KO: Ormdl1For 5'-TGTAATGAACAGCCGTGGTAT-3'; Ormdl1Rev 5'-GCAGAAGGGGATGCTGAGTAATA-3'. PCR conditions: denaturation, 94°C for 10 min; amplification, 94°C for 20 s, 60°

Scientific). Total ceramide and dihydroceramide levels were determined by summing the individual fatty-acid species.

## Western blot

Expression of the fSPT transgene was detected in total tissue lysates. Sciatic nerves were harvested and homogenized in RIPA buffer (ThermoFisher Scientific) supplemented with protease inhibitor cocktail (ThermoFisher Scientific) and phosphatase inhibitor cocktail (Roche Life Science, Indianapolis, IN). Liver was harvested and homogenized in Cell Lysis Buffer (Cat#9803 s, Cell Signaling Technology, Danvers, MA) with protease inhibitors (Crystalgen, Commack, NY). Samples were incubated 30 min on ice and later spun at 14,000 x $g$ at 4°C for 10 min. Proteins were resolved on a 4–12% Bis-Tris gel (ThermoFisher Scientific), then transferred to nitrocellulose membranes (ThermoFisher Scientific). The membranes were blocked in 5% non-fat dry milk, 0.05% Tween-20 in TBS, then probed with the indicated antibodies. fSPT expression was detected using monoclonal anti-human SPTLC1 (clone H-1, sc-374143, 1:1000 dilution) (Santa Cruz Biotechnology, Dallas, TX), followed by anti-mouse IgG, HRP-conjugated (AP-124P, Millipore, Burlington, MA). Membranes were reprobed with anti-mouse β-actin (monoclonal AC-15, HRP-conjugated) (Abcam, Cambridge, MA) to provide a loading control. Signal was detected by chemiluminescence (ECL Prime Western Blotting Detection Reagents, GE Healthcare Bio-Sciences, Pittsburgh, PA) visualized with an Amersham Imager 680 blot and gel imager (GE Healthcare Bio-Sciences).

## SPT activity

SPT enzymatic activities were measured by acyl-CoA dependent incorporation of [$^3$H] serine into sphingoid bases as previously described (*Harmon et al., 2013*). Each assay was performed using liver microsomes (200 μg protein) with 50 μM palmitoyl-CoA.

## Statistical analysis

Sample sizes were estimated on the basis of our previous publications (*Alexaki et al., 2017*; *Alexaki et al., 2014*; *Bektas et al., 2010*; *Taguchi et al., 2016*). Statistical analysis was performed using unpaired Student's $t$ test, one-way analysis of variance (ANOVA) with Bonferroni's multiple comparison test, or Chi-square analysis, as described in the legends, using Prism software (GraphPad, San Diego, CA). P-values <0.05 were considered statistically significant.

## Data availability

All data generated during this study are included in the manuscript and supporting files. Source data files have been provided.

## Acknowledgements

This work was supported by the Intramural Research Program of the National Institutes of Health, the National Institute of Diabetes and Digestive and Kidney Diseases (RLP), a Fondation Leducq transatlantic network grant (SphingoNet; RLP) and the Lipidomics Shared Resource, Hollings Cancer Center, Medical University of South Carolina (P30 CA138313 and P30 GM103339). We thank Ewa Turner for laboratory management. The content is solely the responsibility of the authors and does not necessarily represent the official views of the National Institutes of Health.

## Additional information

### Funding

| Funder | Grant reference number | Author |
|---|---|---|
| Fondation Leducq | SphingoNet | Richard L Proia |
| National Institute of Diabetes and Digestive and Kidney Diseases | Intramural Research Program | Richard L Proia |

The funders had no role in study design, data collection and interpretation, or the decision to submit the work for publication.

## Author contributions

Benjamin A Clarke, Conceptualization, Formal analysis, Investigation, Writing - original draft, Writing - review and editing; Saurav Majumder, Hongling Zhu, Conceptualization, Formal analysis, Investigation, Writing - review and editing; Y Terry Lee, Mari Kono, Cuiling Li, Caroline Khanna, Hailey Blain, Ronit Schwartz, Vienna L Huso, Colleen Byrnes, Galina Tuymetova, Formal analysis, Investigation, Writing - review and editing; Teresa M Dunn, Formal analysis, Methodology, Writing - review and editing; Maria L Allende, Conceptualization, Investigation, Formal analysis, Writing - review and editing; Richard L Proia, Conceptualization, Funding acquisition, Investigation, Supervision, Resources, Project administration

## Author ORCIDs

Mari Kono (iD) https://orcid.org/0000-0003-2447-4350
Caroline Khanna (iD) http://orcid.org/0000-0002-4356-3933
Ronit Schwartz (iD) http://orcid.org/0000-0003-4297-3495
Richard L Proia (iD) https://orcid.org/0000-0003-0456-1270

## Ethics

Animal experimentation: All animal procedures were approved by the National Institute of Diabetes and Digestive and Kidney Diseases Animal Care and Use Committee and were performed in accordance with National Institutes of Health guidelines under protocol number K007-GDDB-18.

## Decision letter and Author response

Decision letter https://doi.org/10.7554/eLife.51067.sa1
Author response https://doi.org/10.7554/eLife.51067.sa2

# Additional files

## Supplementary files

• Transparent reporting form

## Data availability

All data generated or analyzed during this study are included in the manuscript and supporting files.

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
