## [Decision Letter]

**Acceptance summary:**

Sphingolipids play critical roles in many cellular signaling pathways – they form the myelin membrane that insulates neuronal axons and their bioactive metabolites such as ceramide regulate cell growth. Because of their importance and the potential toxicity of metabolic intermediates, sphingolipid levels are carefully regulated by feedback inhibition of its synthesis, a process mediated by the Ormdl family of proteins. The three members of this family sense elevated ceramide levels and inhibit a key enzyme in the sphingolipid biosynthesis pathway to ultimately lower sphingolipid levels. Still, the functional importance of Ormdls in physiological settings remains unclear. Here, Clarke et al. address this comprehensively by generating knockout mice for each of the three Ormdl genes, for combinations of two of the three genes, and for all three genes. Knockout of all three Ormdls is embryonic-lethal, emphasizing the essential nature of this regulation. Moreover, they find that the Ormdls function in an isoform-specific manner to maintain control of sphingolipid biosynthesis in a specific physiological context – myelination – where there is a high demand for these lipids. In the absence of proper Ormdl-mediated regulation, they find elevated levels of sphingolipids and toxic intermediates, which lead to severe myelination defects and neurological abnormalities. These studies bring a deeper understanding of the role of the Ormdl family of proteins in the tight regulation of sphingolipid biosynthesis, an essential feature of all cells, but particularly in tissues such as in the brain, skin, and testes, where high levels of sphingolipid synthesis occur.

**Decision letter after peer review:**

Thank you for submitting your article "The Ormdl genes regulate the sphingolipid synthesis pathway to ensure proper myelination and neurologic function in mice" for consideration by *eLife*. Your article has been reviewed by three peer reviewers, and the evaluation has been overseen by a Reviewing Editor and Huda Zoghbi as the Senior Editor. The following individuals involved in review of your submission have agreed to reveal their identity: Brian Wattenberg (Reviewer #1); Gerhild van Echten-Deckert (Reviewer #2); Roger Sandhoff (Reviewer #3).

The reviewers have discussed the reviews with one another and the Reviewing Editor has drafted this decision to help you prepare a revised submission.

Summary:

This article is an important contribution to the sphingolipid and myelination fields. It presents the first description of the combined deletion of all three ORMDL proteins in mice, which leads to impaired myelination and hence severe neurological problems. ORMDL proteins are the major regulators of sphingolipid synthesis and this paper provides us with an excellent model to study the regulation and function of this important lipid class. All three reviewers were highly enthusiastic about the study and judged it to be of significant interest to scientists working on lipid homeostasis and neuronal function. Like all pioneering studies, there are a few issues raised from the data that are listed below. Addressing these points with data that the authors may already have or by discussion could further enhance the clarity and impact of this study.

Essential revisions:

1) Given the increased amounts of ceramide in the myelin of ORMDL1,3 double knockouts, it would be interesting to know the amount of GalCer and of sulfatide in these samples. This data would enhance the paper, but is not strictly required.

2) The latest study on ORMDLs by Davis et al., 2019, showed that in addition to ceramide, sphingosine could also exert a significant inhibitory effect of ORMDLs on SPT. Further, it has been suggested that in neurons lacking S1P-lyase, the excess of free and phosphorylated sphingoid bases is responsible for ORMDL1,3 inhibition of SPT. No changes in ceramide amounts were detected in those neurons. Thus, Figure 1A might be misleading suggesting that ceramide is the only culprit. This point could be briefly addressed in the text.

3) In the discussion of Morbust Charcot-Marie-Tooth (CMT), please include a reference to Atkinson et al., 2017, which shows that S1P-lyase, a catabolic enzyme of sphingolipid metabolism is also involved in CMT.

4) The authors find no upregulation of (dihydro)ceramides with very long chain fatty acids (C22-26) in the brain of ORMDL1 and 3-DKO (Figure 1—figure supplement 2), whereas in this DKO the highest upregulation of (dihydro)ceramides in the sciatic nerve is found exactly with those containing very long chain fatty acids (Figure 3—figure supplement 1). As the major phenotype described affects the sciatic nerve, it may be worth picking up this result with one or two sentences in the Discussion. Especially the downstream products of ceramides in myelin-producing cells, the cerebrosides and sulfatides are very rich in very long chain fatty acids. Might there be a difference in this downstream flux between oligodendrocytes of the brain and Schwann cells of the sciatic nerve?

5) In the global knockouts, it is not clear if the effects on myelination are on the myelinating cells or, for example, on glial or neuronal cells. This is approached in the manuscript by the experiments with oligodendrocyte/Schwann cell-specific overexpression of SPT. The authors note that effects could be on reduced viability of the myelinating cells, but do not measure that directly. The major sphingolipids in myelin are cerebrosides and sulfatides, and to a lesser extent sphingomyelin. If available, it would be interesting to know the levels of those lipids, which would directly impact on the observation of redundant myelin.

6) Table 1 first line, second to last column states: *Ormdl1^-/-^ Ormdl2^-/-^ Ormdl^+/+^*. However, shouldn't it be *Ormdl1^-/-^ Ormdl2^-/-^ Ormdl^+/-^*?

7) Legend of Figure 1 should contain a link to the supplemental showing which (dihydro)ceramide species underlie the levels presented here.

8) Legend of Figure 4—figure supplement 1, Figure 1—figure supplement 2, Figure 3—figure supplement 1, and figure 4—figure supplement 2 should contain short information as to the length of the sphingoid base analyzed to reveal the levels of sphingoid bases and ceramides (most likely it was the major one with 18 carbons).

---

## [Author Response]

Essential revisions:1) Given the increased amounts of ceramide in the myelin of ORMDL1,3 double knockouts, it would be interesting to know the amount of GalCer and of sulfatide in these samples. This data would enhance the paper, but is not strictly required.

Thank you for this excellent suggestion. We have now included levels of hexosylceramide, comprising both glucosyl and galactosylceramide, in sciatic nerves from *Ormdl1* single KO, *Ormdl3* single KO, *Ormdl1/3* double KO and WT mice (Figure 3E). Hexosylceramide levels in the *Ormdl1* and *Ormdl3* single KO sciatic nerves were increased by about 20-50% compared to WT. In sciatic nerves from *Ormdl1/3* double KO mice, hexosylceramide levels were lower by about 50% compared to WT, even though they contained 14 times dihydroceramide and the amount 8 times the amount of ceramide of the WT (Figure 3E).

These results are in line with the concept outlined in the Discussion (second paragraph) in which ORMDLs are necessary to restrain SPT activity to insure that the amounts of precursor sphingolipids produced do not exceed the capacity of the pathway to generate mature sphingolipids (hexosylceramides). Accordingly, in the absence of both ORMDL 1 and 3, the amounts of precursor ceramide produced greatly exceed the capacity the terminal steps of the pathway to convert it to mature hexosylceramide. The buildup of excessively high sphingolipid precursor levels in the *Ormdl* 1/3 double KO sciatic nerve may cause cellular damage resulting in the loss of myelinated axons, and, thus, the reduced hexosylceramide levels.

2) The latest study on ORMDLs by Davis et al., 2019, showed that in addition to ceramide, sphingosine could also exert a significant inhibitory effect of ORMDLs on SPT. Further, it has been suggested that in neurons lacking S1P-lyase, the excess of free and phosphorylated sphingoid bases is responsible for ORMDL1,3 inhibition of SPT. No changes in ceramide amounts were detected in those neurons. Thus, Figure 1A might be misleading suggesting that ceramide is the only culprit. This point could be briefly addressed in the text.

The possible involvement of free and phosphorylated sphingoid bases in the ORMDL-mediated SPT regulation is now included in the Discussion (first paragraph).

3) In the discussion of Morbust Charcot-Marie-Tooth (CMT), please include a reference to Atkinson et al., 2017, which shows that S1P-lyase, a catabolic enzyme of sphingolipid metabolism is also involved in CMT.

We have included this interesting point and reference in the Discussion (third paragraph).

4) The authors find no upregulation of (dihydro)ceramides with very long chain fatty acids (C22-26) in the brain of ORMDL1 and 3-DKO (Figure 1—figure supplement 2), whereas in this DKO the highest upregulation of (dihydro)ceramides in the sciatic nerve is found exactly with those containing very long chain fatty acids (Figure 3—figure supplement 1). As the major phenotype described affects the sciatic nerve, it may be worth picking up this result with one or two sentences in the Discussion. Especially the downstream products of ceramides in myelin-producing cells, the cerebrosides and sulfatides are very rich in very long chain fatty acids. Might there be a difference in this downstream flux between oligodendrocytes of the brain and Schwann cells of the sciatic nerve?

We have now addressed this point in the Discussion (fifth paragraph).

5) In the global knockouts, it is not clear if the effects on myelination are on the myelinating cells or, for example, on glial or neuronal cells. This is approached in the manuscript by the experiments with oligodendrocyte/Schwann cell-specific overexpression of SPT. The authors note that effects could be on reduced viability of the myelinating cells, but do not measure that directly. The major sphingolipids in myelin are cerebrosides and sulfatides, and to a lesser extent sphingomyelin. If available, it would be interesting to know the levels of those lipids, which would directly impact on the observation of redundant myelin.

This is a good point. As discussed in the response to comment 1, we have now included measurements of hexosylceramide levels of sciatic nerves from *Ormdl1* single KO, *Ormdl3* single KO, *Ormdl1/3* double KO and WT mice (Figure 3E).

We have also included the hexosylceramide levels for sciatic nerve with oligodendrocyte/Schwann cell-specific overexpression of SPT (Figure 4G). Similar to the *Ormdl*1/3 double KO mice, we find levels amount of lower levels hexosylceramide but extremely high levels of dihydroceramide and ceramide compared with WT. The reduced levels of hexosylceramide can be attributed to loss of myelinated axons.

The abnormally high ceramide level relative to the hexosylceramide level suggests that abnormal myelin lipid composition due to excessive, unbalanced sphingolipid synthesis may be a cause of redundant myelination, as outlined in the Discussion.

6) Table 1 first line, second to last column states: Ormdl1^-/-^ Ormdl2^-/-^ Ormdl^+/+^. However, shouldn't it be Ormdl1^-/-^ Ormdl2^-/-^ Ormdl^+/-^?

Thank you for pointing this out. The top row of Table 1 has been corrected and the data has been updated.

7) Legend of Figure 1 should contain a link to the supplemental showing which (dihydro)ceramide species underlie the levels presented here.

Supplementary figures have now been included as figure supplements in all cases.

8) Legend of Figure 4—figure supplement 1, Figure 1—figure supplement 2, Figure 3—figure supplement 1, and Figure 4—figure supplement 2 should contain short information as to the length of the sphingoid base analyzed to reveal the levels of sphingoid bases and ceramides (most likely it was the major one with 18 carbons).

The length of the sphingoid base in all the determinations was C18. This information has been added to the legends of Figure 1—figure supplement 2, Figure 3—figure supplement 1, Figure 4—figure supplement 1, Figure 4—figure supplement 2.